# Transversus Abdominis Plane (TAP) Block in Rabbit Cadavers: Anatomical Description and Measurements of Injectate Spread Using One- and Two-Point Approaches

**DOI:** 10.3390/ani14050684

**Published:** 2024-02-22

**Authors:** Federica Serino, Luca Pennasilico, Margherita Galosi, Angela Palumbo Piccionello, Adolfo Maria Tambella, Caterina Di Bella

**Affiliations:** School of Bioscience and Veterinary Medicine, University of Camerino, 62024 Matelica, Italy; federica.serino@unicam.it (F.S.); luca.pennasilico@unicam.it (L.P.); margherita.galosi@unicam.it (M.G.); angela.palumbo@unicam.it (A.P.P.); adolfomaria.tambella@unicam.it (A.M.T.)

**Keywords:** regional anesthesia, rabbit, transversus abdominis plane, spread, lidocaine

## Abstract

**Simple Summary:**

Anesthetic risk in rabbits is higher than in other species such as dogs and cats and, furthermore, the treatment of their perioperative pain is sometimes inadequate. Loco-regional anesthesia is seeing increasing application in veterinary medicine, as it allows a reduction in the systemic administration of drugs. The block of the transversus abdominis plane (TAP) is an anesthetic technique of great interest for locoregional anesthesia in veterinary practice. This technique is used to desensitize the abdominal wall and consists of injecting a volume of local anesthetic into the TAP with the aim of obtaining its distribution on the branches of the thoracolumbar nerve located in the fascial plane between the internal oblique and transversus abdominis muscles. In both human and veterinary medicine, this technique is considered promising for producing analgesia for surgical procedures performed on the abdominal wall (e.g., laparoscopies, mastectomies, abdominal hernias), representing a valid alternative to epidural analgesia. Different approaches to performing ultrasound-guided TAP block have been described. The preiliac approach is one of the most used; however, it seems to be useful in managing pain in the caudal abdomen, but the cranial diffusion of the local anesthetic does not seem to be sufficient to achieve complete desensitization of the abdominal wall. Alternative approaches, involving further retrocostal and/or subcostal infiltration, have been successfully studied in dogs, cats, calves and ponies to ensure effective desensitization of the cranial abdomen and, in part, also of the thorax; but these techniques do not have yet been described in rabbits.

**Abstract:**

The aim of this study was to describe one-point (preiliac approach) and two-point (preiliac and retrocostal approach) blocks of the Transversus Abdominis Plane (TAP) on a cadaveric model. For this purpose, ultrasound-guided infiltration of the plane between the internal oblique and transversus abdominis muscles was performed and, after dissection of tissues, the areas and percentage of nerve fibers involved were analyzed. Injection into the TAP plexus of a 1 mL/kg solution of 2% lidocaine and 1% methylene blue (1:1) was performed in 30 rabbit cadavers. In fifteen rabbits (group S), the solution was inoculated at the preiliac level. In the other 15 rabbits (group D), the solution was divided into two inoculations (0.5 mL/kg at the retrocostal level and 0.5 mL/kg at the preiliac level). All cadavers were then dissected and stained spinal nerve branches were measured. Moreover, the percentage of length, height and the total area of the stained tissue were calculated. In the S group, T10, T11 and T12 nerve eminences were successfully stained in 18%, 52% and 75% of cases, respectively. Furthermore, L1, L2, L3 and L4 were stained in 95%, 100%, 60% and 40% of cases, respectively. In group D, the ventromedial eminence of T10, T11 and T12 were stained in 68.1%, 100% and 98% of cases, respectively, and L1, L2, L3 and L4 were stained in 88%, 100%, 62% and 31% of cases, respectively. In conclusion, a two-point TAP block is more effective in covering the nerve eminences of the cranial abdomen than the preiliac approach alone.

## 1. Introduction

Rabbits, compared to other companion animals, require careful anesthetic management [1]. In this species, some physiological characteristics must be considered, such as a fast metabolism, which predisposes these animals to hypoglycemia and hypoxia, and the difficult orotracheal intubation due to the conformation of the oral cavity and the larynx (the presence of long incisor teeth, a large tongue and a deep larynx) [2,3]. Furthermore, these animals are susceptible to high levels of stress due to handling or painful stimulation, resulting in the release of endogenous catecholamines, potentially lethal cardiac arrhythmias, loss of appetite, reduced peristalsis and hepatic lipidosis. For these reasons, a good perioperative analgesic treatment is one of the most important aspects to consider during surgical procedures, in order to reduce the nociceptive stimuli, ensure a better recovery and reduce morbidity and mortality in this species [4,5]. Currently, the use of locoregional anesthesia is a valid analgesic choice, with the aim of reducing the administration of opioids, as well as the requesting of halogenated anesthetics, reducing side effects of systemic drugs and achieving a faster recovery [6,7]. Transversus Abdominis Plane block (TAP block) is a locoregional anesthesia technique widely used in human medicine and, recently, also in veterinary practice, in order to manage the perioperative pain induced by specific types of abdominal surgery (e.g., laparoscopies, mastectomies, abdominal hernias and cesarean delivery), representing a valid alternative to epidural analgesia [8,9]. TAP block consists of desensitization of the abdominal wall. In this region, the lateral abdominal muscles consist of three layers: the external oblique (EO), internal oblique (IO) and transversus abdominis (TA) muscles. Previous studies performed in a dog cadaveric model showed that the ventral branches of the nerves T9–T13 and L1–L3 run in the fascia between the transversus abdominis and the internal oblique muscle. The technique involves infiltrating a local anesthetic (LA) in the fascial plane between the transversus abdominis and the internal oblique muscles (TAP plexus), under ultrasound guidance, in order to block sensory transmission from the afferent pathways of the ventral thoracolumbar nerve eminences and, therefore, desensitize the skin, subcutis, muscle planes and parietal peritoneum of the abdominal wall [7,8,9,10]. Both in human and veterinary medicine, several approaches to performing ultrasound-guided (US-guided) TAP block have been described. Specifically, the preiliac approach is one of the most frequently used. This technique involves positioning the patient in lateral recumbency while the US probe is placed in a transverse orientation to the spine, midway between the iliac crest and the last rib, lateral to the midline of the abdomen. This approach appears to be useful in the pain management of the caudal abdomen; however, the cranial spread of the LA does not appear to be sufficient to achieve the complete desensitization of the cranial abdominal wall [11]. Previous studies in human medicine have already shown that the use of the multi-point TAP block can guarantee a greater cranial and caudal spread compared to the single-point approach [12]. Subsequently, thanks to recent studies applied to various animal species, the hypothesis that the preiliac technique alone is not sufficient was also confirmed in veterinary medicine. Indeed, alternative approaches, involving additional retrocostal and/or subcostal infiltration, have been successfully studied in species as dogs, cats and calves to ensure effective desensitization of the cranial abdomen and, in part, also of the thorax [13,14,15]. Recently, Otero et al. also hypothesized that the three-point TAP (retrocostal, subcostal and preiliac approaches) produces a more consistent distribution of the injectate compared to a two-point injection (subcostal and preiliac blocks), with greater involvement of the ventromedial branches from T10 to L2 and an increase in the staining rate on T13 and L1, in cat cadavers [14]. To the authors’ knowledge, there is only one reference describing TAP block in a rabbit cadaver and there are no relevant in vivo clinical studies. Furthermore, in the aforementioned study, only the one-point TAP block was performed (preiliac approach). The authors themselves state that this technique is probably not sufficient to achieve complete desensitization of the rabbit abdomen and that further research is needed to evaluate the best approach to use in this species [16]. Based on this background, the aim of this study was to describe two US-guided TAP block techniques in rabbits and to compare the one-point (preiliac) and two-point (preiliac and retrocostal) techniques in a cadaveric model of a rabbit. For this purpose, the ultrasound-guided infiltration in the TAP was performed using lidocaine 2% and methylene blue 1% (1:1 solution). After the block, the anatomic dissection was performed in order to measure and compare the infiltrated areas and the number of stained nerves. Our hypothesis is that the two-point approach involves a larger area of the abdomen, staining a greater number of nerves compared to the one-point approach.

## 2. Materials and Methods 

This study included a total of 30 adult New Zealand rabbit cadavers. The animals were not euthanized for the purposes of the study. Furthermore, all cadavers came from affiliated breeding and were free from transmissible infectious diseases. Based on this, it was not necessary to request permission from the ethics commission. The subjects, which were dead for no more than 12 h, were immediately frozen and thawed at room temperature 48 h before the procedure. The degree of thawing was confirmed by the operator on each cadaver before the study began, and animals with abdominal alterations were excluded. 

All subjects were divided into two groups, including hemiabdomen, in the study:S group (15 rabbits; 30 hemiabdomen) in which the one-point (preiliac) approach was carried out;D group = (15 rabbits; 30 hemiabdomen) in which the two-point approach was carried out (preiliac and retrocostal).

The study was therefore divided into two phases: (1) ultrasound-guided injection of the TAP; (2) anatomical dissection and measurement of stained areas and nerve eminences on all hemiabdomen.

### 2.1. US-Guided TAP Injection

In both groups, the rabbits were placed in lateral recumbency and the hair was clipped from the first rib to the sacrum, bilaterally. All ultrasonographic images were acquired by the same operator using a specific US system (MyLab 9; Esaote Spa; Genova, Italy). In S group, linear array probe (13–6 MHz; L3–11 probe; Esaote Spa; Genova, Italy) was positioned transversely to the spine, in the focus of an imaginary line starting from the iliac crest and ending at the caudal margin of the last rib [17]. US image was deemed suitable if the three muscle planes (external oblique, EO; internal oblique, IO; transversus abdominis, TA), peritoneum and intraperitoneal structures were visible (Figure 1).

After calculating the thickness of the TAP, a prefilled 22-gauge, 33 mm echogenic needle (Stimuplex Ultra 360 22G; B. Braun; Melsungen, Germany) was introduced in a dorsal-to-ventral direction between the muscle layers using an in-plane approach. When the fascial plane between the IO and TA was reached, 0.2 mL of 0.9% NaCl was inoculated to confirm correct needle placement. The appearance of irregular opacity could mean intramuscular inoculation; thus, the needle was repositioned to perform a proper inoculation. Any incorrect placement was recorded. Then, once the needle was correctly positioned, it was connected to an extension and 1 mL/kg of 2% lidocaine (Ecuphar Italia S.r.l., Milan, Italy) mixed with 1% methylene blue (Alcyon Spa; Rome, Italy) as a 1:1 solution was inoculated. During the inoculation, it was possible to observe the hydrodissection of the muscle planes, which appeared as a hypoechoic space (Figure 2).

After holding the position for 5 min, the same procedure was performed on the opposite hemiabdomen. In group D, in addition to performing the previously described technique, the TAP plexus was localized at the retrocostal level. The bodies were placed in dorsal recumbency, and the probe was positioned parallel to the last rib. The rectus abdominis muscle was identified as reference point, therefore, the probe was moved in caudolateral direction until a good visualization of the TAP plane was obtained (Figure 3). The needle was, then, inserted in-plane in a ventral-to-dorsal direction, and the inoculation was performed as previously described. In group D, the dose of lidocaine/methylene blue (1:1 solution) used was 0.5 mL/kg per site.

### 2.2. Anatomical Dissection and Misurements

Anatomical dissection of cadavers was carried out to assess the distribution of the inoculated solution and to quantify the number and percentage of stained nerves. It was performed 60 min after the US-guided blocks. Before starting, the length of the vertebral column from the last rib to the iliac crest (L_abd_; mm) and the height of the hemiabdomen from the transverse processes of the spine to the linea alba (H_abd_; mm) were measured. Moreover, the body surface area was calculated using a specific formula adapted for rabbit’s body [18]:BSA = 11.0 × BW (kg)^2/3^
(1)

Afterwards, the cadavers were positioned in dorsal recumbency and, through an incision along the midline (from the manubrium of the sternum to the pubis), the skin and subcutaneous tissues were removed, obtaining visualization of the OE and the rectus abdominis muscles. Then, the bodies were moved to lateral recumbency, and the OE and OI muscles were carefully dissected, allowing us to visualize the fascial plane between the latter and the transversus muscle, and identifying the stained areas (Figure 4).

The correct positioning in the fascia and the diffusion of the dye were analyzed by an experienced surgeon and anesthetist. Nerves were considered appropriately stained if the solution was distributed around the nerve over a length > 1 cm [19]. Dye spread was measured in both cranio-caudal (CC_spread_; mm) and dorsal-ventral (DV_spread_; mm) directions, using photos of the areas and applying a reference meter (ruler) always positioned at the height of the spine (Figure 5).

Specifically, the images were transferred into a specific software (ImageJ 1.45 s freeware, National Institutes of Health, Rockville, MD, USA) [20]. The measurement corresponding to 1 mm was standardized using the ruler in the images and it was then correlated to the areas to be analyzed. The height and length of the stained tissues were then measured, calculating the total stained area in both groups (AS and AD, respectively; mm^2^) and relating it to the total BSA (Stained Area/BSA, %). Moreover, the percentage of craniocaudal and dorsoventral spread (L_spread_, %; H_spread_, %, respectively), and the percentage of nerve eminences stained were also calculated. All cases of abdominal puncture, incorrect execution of the blocks or incorrect dissection were recorded.

### 2.3. Statistical Analysis

Statistical analysis was performed using MedCalc software 9.0 (MedCalc version 9.2.10; MedCalc Software). Data were tested for normality with Shapiro–Wilk test and were summarized as mean ± standard deviation. Cardinal variables were analyzed with One-way ANOVA test, to perform a comparison between groups. Instead, for yes/no variables (nerve eminences stained), Fisher’s exact test was used. A *p* value < 0.05 was considered statistically significant.

## 3. Results

Sixty hemiabdomen of New Zealand rabbits (twenty females and ten males) were included in this study (S group = 30; D group = 30). There were no statistically significant differences in weight (S = 3.1 ± 0.2 kg; D = 3.07 ± 0.12 kg), age (S = 13.13 ± 4 months; D = 13.07 ± 3.5 months), Labd (S = 100.49 ± 7.5 mm; D = 112.001 ± 5.42 mm), Habd (S = 85.12 ± 4.2 mm; D = 87.34 ± 5.33 mm) and BSA (S = 245,112.4 ± 11,897.01 mm^2^; D = 237,892.3 ± 20,112.02 mm^2^). This scientific paper conformed to the Consolidated Standards of Reporting Trials (CONSORT) Statement 2010 for reporting randomized clinical trials [21] (Figure 6).

### 3.1. US-Guided TAP Injection

Based on the landmarks used and previously described, we were able to obtain a good visualization of the TAP in all hemiabdomen. It was difficult for the operators to clearly visualize the muscle layers of the right hemiabdomen due to the presence of the cecum-colon and the artefacts caused by the meteorism. In D group, two hemiabdomen were excluded, due to intraperitoneal infiltration during the injection at the preiliac level, while the retrocostal inoculations were all performed correctly. In S group, the injection point was wrong in five hemiabdomen but, after repositioning the needle, the procedure was performed correctly, and no cases were excluded. Regarding the thickness of the three muscle layers, no statistically significant differences were found at the retrocostal level (EO = 1.2 ± 0.3 mm; IO = 1.35 ± 0.1 mm; TA = 1.3 ± 0.1 mm) compared to the preiliac (EO = 1.2 ± 0.2 mm; IO = 1.4 ± 0.3 mm; TA = 1.35 ± 0.12 mm). 

### 3.2. Anatomical Dissection and Measurement

Cadaveric dissection was performed successfully in all cadavers and the dye was correctly localized between IO and TA. Cranio-caudal diffusion of methylene blue (CC_spread_; mm) was shown to be significant in D (D = 69.36 ± 10 mm) compared to S group (S = 48.91 ± 7.24 mm), while dorso-ventral spread (DV_spread_; mm) was significantly greater in a single-point infiltration (S = 54.98 ± 8.8 mm; D = 35.36 ± 15.1 mm). Moreover, analysis of the stained areas showed that there were no statistically significant differences between the two groups in the study (AS = 2265.551 ± 461.177 mm^2^; AD = 2306.433 ± 656.949 mm^2^) (Table 1).

In S group, cranial spread of methylene blue was detected up to the T11 nerve eminence, which was successfully stained in 52% of cases. T12, L1 and L2 were successfully stained in 75%, 95% and 100% of cases, respectively. L3 and L4 nerve eminences were successfully stained in 60% and 40% of cases, respectively. However, no cranial nerves to T11 and caudal to L4 were stained. In D group, T10 was stained in 68% of cases and T11 and T12 were stained in 100% and 98% of cases, respectively. Regarding the lumbar eminences, L1, L2, L3 and L4 were stained in 88%, 100%, 62% and 31% of cases, respectively (Table 2). 

## 4. Discussion

TAP block is a loco-regional technique widely used in human and veterinary medicine. However, to date, there are no studies describing and comparing TAP block techniques at one and two points in rabbits. In a previous cadaveric study, it was demonstrated that the preiliac approach alone is not sufficient to obtain good desensitization of the cranial abdomen, and this technique is probably more suitable for surgeries involving exclusively the caudal part of the abdomen [16,22]. The results obtained in this study showed that the combination of a preiliac and retrocostal approach could be a better technique for achieving complete abdominal wall block in rabbits, guaranteeing a greater and more complete desensitization of tissues, compared to the application of the single pre-iliac TAP block.

Retrocostal ultrasound visualization was performed successfully in all cadavers without difficulty. However, in the preiliac approach, two hemiabdomens were inoculated incorrectly and, in five hemiabdomens, the needle had to be repositioned. Probably the greatest difficulty encountered in performing the preiliac block was due to the different anatomy of the rabbit compared to dogs and cats. In fact, rabbits have a voluminous cecum-colon, detectable by ultrasonography, at the level of the right hemiabdomen, in contact with the abdominal wall. On a cadaveric model, the presence of fermentations and meteorism creates artifacts that make the ultrasound visualization of the three muscle planes more complex [23,24]. We hypothesize, therefore, that visualization of the abdominal wall on live rabbits should be better. As regards the thickness of the three muscular planes involved in the TAP block, in this study, it was highlighted that there are no differences in the thickness of the three planes at the preiliac level compared to the retrocostal position and, therefore, this aspect did not influence the execution of the block, nor the correct dissection of the planes. Muscle layers that are too thin could make the procedure more difficult. Otero et al. excluded abdomens with a muscle thickness less than 2.5 mm. In agreement with them, it is our opinion that this technique is not recommended in subjects with an abdominal muscle wall thickness of less than 2.5 mm [14]. 

Concerning the results obtained from the measurements of the stained areas, we highlighted that in group D, the caudo-cranial diffusion is greater than in group S, while the dorsal–ventral diffusion is greater in the latter. The authors’ opinion is that the division of the same volume into two different inoculations allowed a greater cranial distribution of methylene blue. On the contrary, by placing a large volume in a single point, it is distributed uniformly in the caudocranial and ventral–dorsal direction, accumulating in situ. Our study agrees with what was described by Freitag et al., who demonstrated that high volumes of local anesthetic are not necessary to achieve greater caudo-cranial diffusion and that the analgesic efficacy of a single high-volume block is not better than that of a two-point block at lower volumes [11]. To further confirm this, the stained area in group S is not significantly different compared to the summation of the areas of the two-point block. This is probably due to the prevailing distribution of the dye in the caudocranial direction in D group, and, on the contrary, the equal cranio-caudal and dorso-ventral distribution of fluid in S group. According to our work, previous studies performed in dog cadavers showed that the inoculation of a large volume of dye (i.e., 1 mL/kg) into the TAP plane tended to accumulate at the injection site, while inoculation of a smaller volume at different points resulted in reliable staining of the most cranial nerves [25,26]. Furthermore, the analysis of nerve eminences involved showed that, in rabbits, the single-point TAP block is not able to guarantee complete desensitization of the cranial abdomen. On the contrary, according to what was demonstrated in dogs and cats, the association of a second cranial (retrocostal) TAP block would guarantee complete analgesia of the entire abdomen. Specifically, in D group, the results showed the involvement of the nerve eminences up to T10 (cranially) and up to L3–L4 (caudally) [6,14,27]. The spread of the injected volume and the characteristics of these two different approaches were evaluated in dogs, cats, ponies, calves, and in a Canada lynx, in vivo and in cadaveric studies and, currently, the available literature supports the data obtained in our study [8,14,15,28,29,30].

A limitation to consider in this study is that the diffusion of the local anesthetic solution in the cadaveric model is different compared to that observed in live subjects. The fascial planes in living animals exhibit contractile capabilities that result in active transport of anesthetic fluid [30] We therefore hypothesize that the local anesthetic solution would spread more in these. Therefore, it is necessary to perform in vivo clinical studies comparing the two-point and single-point approaches, in order to evaluate the real clinical application of the two loco-regional techniques in rabbits. Furthermore, in this study, we chose moderate/high volumes of local anesthetic and did not seek the minimum effective dose in this species. This aspect will certainly need to be considered in future clinical studies, considering the toxic dose of lidocaine in rabbits. Another aspect that the authors want to underline is that, as previously demonstrated in human medicine, the TAP block does not guarantee visceral analgesia [31]. For this reason, this technique can be an excellent adjuvant in abdominal surgeries (e.g., oophorectomies, hepatic lobectomies, laparoscopies), where it can be associated with other loco-regional procedures or drugs administered systemically [32]. On the other hand, its analgesic action is successfully carried out in surgeries such as mastectomies, abdominal hernias and exeresis of superficial masses [6,33].

## 5. Conclusions

In conclusion, the results obtained in our study could be very useful for choosing the correct loco-regional technique to apply in rabbits undergoing surgery involving both the cranial and caudal abdomen. Specifically, the two-point TAP block could ensure greater cranial spread of the local anesthetic, thus obtaining better analgesia. If our results are confirmed by in vivo clinical studies, it could allow us to reduce the use of systemically administered opioids, thus limiting side effects (e.g., slowing of peristalsis, loss of appetite, sensory depression), which are particularly relevant in rabbits and rodents.

## Figures and Tables

**Figure 1 animals-14-00684-f001:**
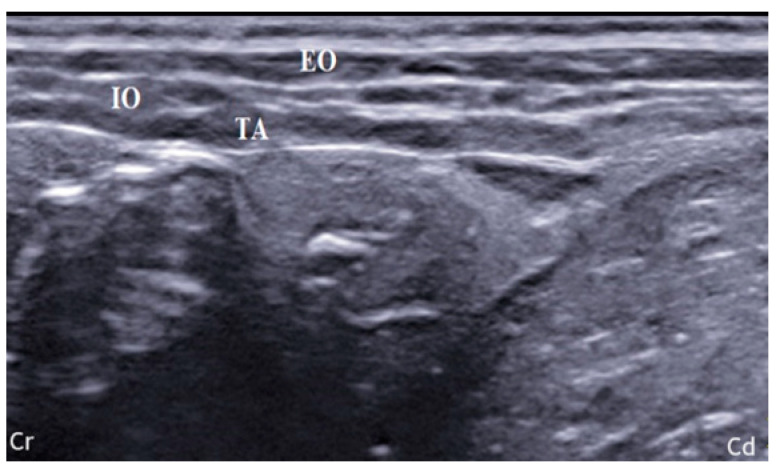
Ultrasonographic visualization of the TAP plane in the preiliac position of the probe. EO: external oblique muscle; IO: internal oblique muscle; TA: transversus abdominis muscle; Cr, Cd: cranial and caudal orientation, respectively; ref. [16].

**Figure 2 animals-14-00684-f002:**
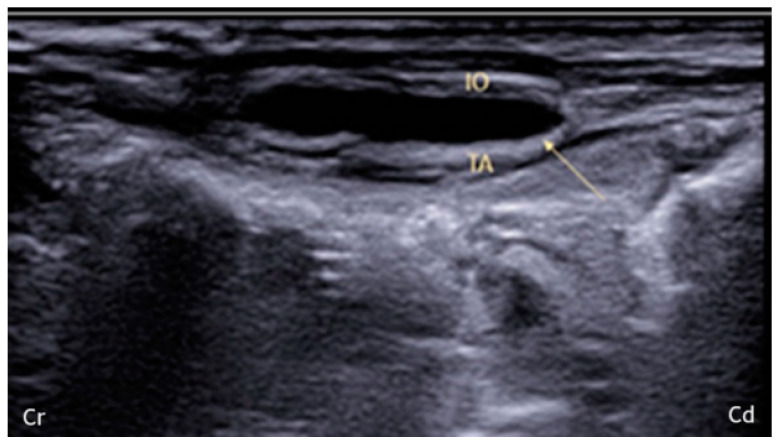
Ultrasonographic visualization of the spread after fascial infiltration. The arrow points to a hypoechoic space between the muscle planes, corresponding to hydrodissection (i.e., separation of the muscular layers) within the transversus fascia after the injection. IO: internal oblique; TA: transversus abdominis muscle; Cr, Cd: cranial and caudal orientation, respectively; ref. [16].

**Figure 3 animals-14-00684-f003:**
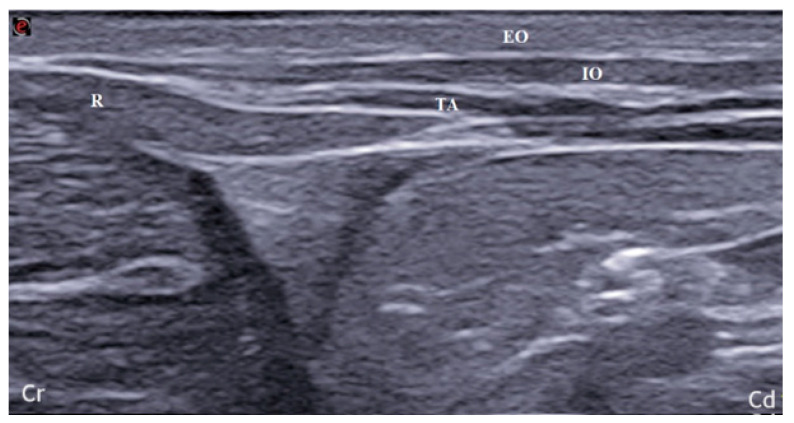
Ultrasonographic visualization of the TAP plane in the retrocostal position of the probe; R: rectus abdominis; EO: external oblique; IO: internal oblique; TA: transversus abdominis muscle; Cr, Cd: cranial and caudal orientation, respectively.

**Figure 4 animals-14-00684-f004:**
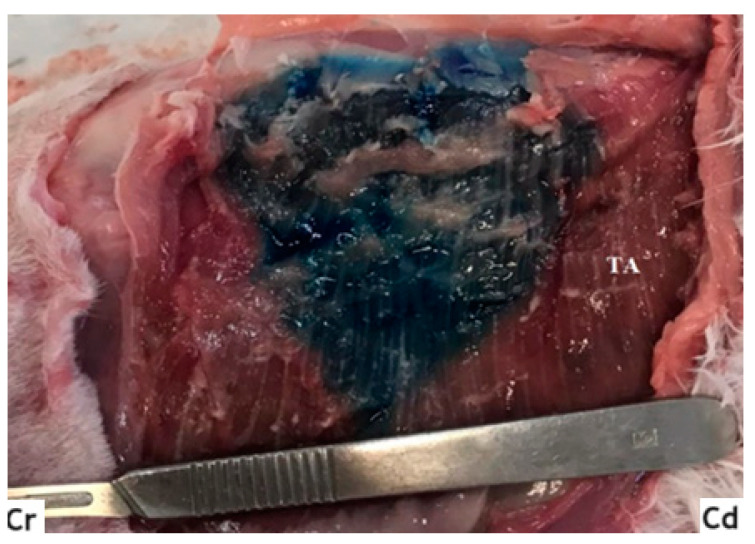
Stained area after the TAP block (one-point technique). TA: transversus abdominis muscle; Cr, Cd: cranial and caudal orientation, respectively; ref. [16].

**Figure 5 animals-14-00684-f005:**
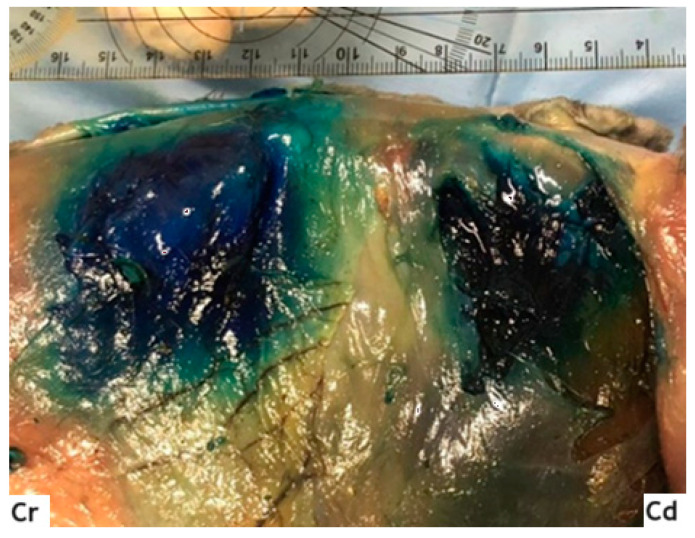
Anatomical dissection after two-point infiltration. Measurements were taken using photos including a reference meter. Cr, Cd: cranial and caudal orientation, respectively.

**Figure 6 animals-14-00684-f006:**
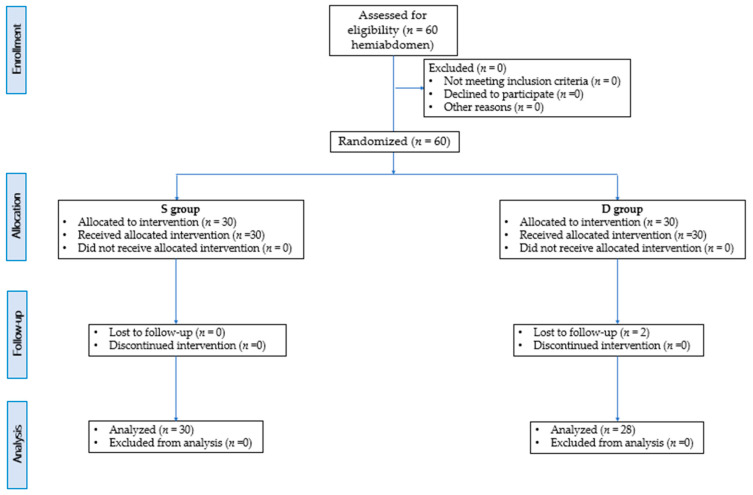
Consolidated Standards of Reporting Trials (CONSORT) flow diagram for hemiabdomens included in the study.

**Table 1 animals-14-00684-t001:** Mean ± SD of the main measurements performed in the two groups of the study. CC_spread_ = caudo-cranial spread of methylene blue; DV_spread_ = dorso-ventral spread of methylene blue; area spread = total strained area. * *p* < 0.05 (statistically significant differences between the two groups of the study).

Group	CC_spread_ (mm)	DV_spread_ (mm)	Area Spread (mm^2^)
S	48.91 ± 7.24	54.98 ± 8.8	2265.5 ± 461.17
D	69.36 ± 10 *	35.36 ± 7.12 *	2306.4 ± 656.9

**Table 2 animals-14-00684-t002:** Percentage of the ventral nerve branches stained by methylene blue, from thoracic (T10) to lumbar (L4) eminences, in the two groups of the study. * *p* < 0.05 (statistically significant differences between the two groups of the study).

Group	T10	T11	T12	L1	L2	L3	L4
S	-	52%	75%	95%	100%	60%	40%
D	68% *	100% *	98% *	88%	100%	62%	31%

## Data Availability

The data presented in this study are available on request from the corresponding author.

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
