# Peer review of "Transversus Abdominis Plane (TAP) Block in Rabbit Cadavers: Anatomical Description and Measurements of Injectate Spread Using One- and Two-Point Approaches"

_animals, 2024, doi:10.3390/ani14050684_

Round 1
Reviewer 1 Report
Comments and Suggestions for Authors
Summary Comments:
The author addresses an interesting clinical question regarding the volume spread of local anesthetic will sufficiently cover the abdominal walls with two different approaches of TAP block in rabbits, and describes a well-planned study. Thank you for the opportunity to review the manuscript. Overall I have very few comments.
General Comments:
While there is a growing appreciation for the critical role that loco-regional anesthesia can play in an overall multimodal anesthetic strategy in veterinary practice, the specific usefulness of the TAP block in rabbits is not clearly documented. This study provides a significant contribution to the current knowledge gap regarding the feasibility of performing a TAP block by utilizing a US-guided technique in rabbits. In addition, it further provides information that a preiliac and retrocostal approach could be a better option to achieve complete abdominal wall block in rabbits. However, further research and documentation are needed to evaluate the effectiveness and safety of the TAP block in rabbits for intra- and postoperative analgesia.
Specific comments:
Line 45-52: I appreciate the background review of the higher mortality rate in rabbits compared to dogs and cats. However, this is irrelevant to the knowledge gaps of locoregional techniques in rabbits. I would recommend making the introduction more succinct and relevant.
Line 86: This is a cadaver study, I would suggest perhaps using “distribution of dye solution” to replace “efficacy”
Line 97-98: The author states “animal died naturally” Please clarify that were the causes of death identified before the inclusion? If rabbits died naturally, I would assume the rabbits were old and might have certain degree of muscle wasting. However, the study population is young with age (S=13.13±4months;D= 209 13.07 ± 3.5 months), it seems unusual that they would died naturally.
Line 180: You do not have a reference for the length of the axon that needs to be exposed to local anesthetic to ensure impulse blocking from local anesthetics. Perhaps could have this one to include: Raymond S.A., Steffensen S.C., Gugino L.D. & Strichartz G.R. (1989) The role of length of nerve exposed to local anesthetics in impulse blocking action. Anesthesia and Analgesia 68, 563–570.
Line 276-280: Please rephrase this paragraph, it is confusing what the author trying to express and/or compare the results from 2 different studies and species. Was the muscle thickness affecting the anatomy recognition under ultrasound-guided technique?
Line 319-322: The authors demonstrated that a 2-point TAP block could provide an efficient coverage of the cranial and caudal abdomen. However, this is a cadaver study, whether TAP block could facilitate the overall reduction of systemic opioid use will require further clinical studies to determine the clinical effectiveness of this technique in rabbits. I would recommend only providing the conclusion of your results and leaving the reader to decide the possible clinical effectiveness before we have more evidence in clinical efficacy in rabbits.
Comments on the Quality of English Language
The manuscript is overall well constructive with good scientific writing.
Author Response
Dear reviewer, thanks for your appreciation and for the time dedicated to reviewing our manuscript. You can read the replies to your comments in the attached file.

Reviewer 2 Report
Comments and Suggestions for Authors
The authors compare the efficacy of one-point (preiliac approach) and two-point (preiliac and retrocostal approach) blocks of the Transversus Abdominis Plane (TAP) using a cadaveric rabbit model. The study investigates the spread of an injected solution (a mixture of lidocaine and methylene blue) within the TAP to determine how effectively it covers the nerve branches associated with the abdominal wall.
The methodology described, involving the dissection and analysis of the dyed nerve fibers, is detailed and seems well-structured for the objectives of the study.
The results indicate a clear difference in the coverage of nerve branches between the single-point and two-point approaches, with the two-point approach showing significantly better coverage of targeted nerve branches.
The study expands on the one previously conducted by the authors on rabbits analyzing the application of a single TAP block point: Di Bella et al. (2021) Animals, 11(7):1953. Ultrasound-Guided Lateral Transversus Abdominis Plane (TAP) Block in Rabbits: A Cadaveric Study.
As a result of this new study, it is shown that the two-point TAP block (lateral and subcostal) could represent a better option, particularly when large surgical incisions are required. However, as the authors themselves comment, further studies are necessary to investigate the analgesic effect of these approaches in a clinical setting.
It is especially critical to note that allegedly the TAP block does not provide visceral analgesia, as previously described in human medicine: Ä°pek CB, Kara D, Yılmaz S, YeÅŸiltaÅŸ S, Esen A, Dooply SSSL, Karaaslan K, Türköz A. Comparison of ultrasound-guided transversus abdominis plane block, quadratus lumborum block, and caudal epidural block for perioperative analgesia in pediatric lower abdominal surgery. Turk J Med Sci. 2019 Oct 24;49(5):1395-1402.
This should be discussed by the authors.
Another aspect that the authors should discuss is related to the fact that authors in other species like dogs have used volumes of 0.25 or 0.16 mL kg-1: In this case, a much higher volume is used, 0.5 mL/kg at the retrocostal level and 0.5 mL/kg at the preiliac level, which could even result in toxicity. Doesn't this adulterate the results of the study?
Minor: In the abstract, a brief introduction about the significance of TAP blocks in medical practice and why this study's comparison is crucial could provide better context.
Author Response
Dear reviewer, thanks for taking the time to review our manuscript. You can read the responses to your comments in the attached file.
